# Parallel Differentiation and Plastic Adjustment of Leaf Anatomy in Alpine *Arabidopsis arenosa* Ecotypes

**DOI:** 10.3390/plants11192626

**Published:** 2022-10-06

**Authors:** Clara Bertel, Dominik Kaplenig, Maria Ralser, Erwann Arc, Filip Kolář, Guillaume Wos, Karl Hülber, Andreas Holzinger, Ilse Kranner, Gilbert Neuner

**Affiliations:** 1Department of Botany, University of Innsbruck, 6020 Innsbruck, Austria; 2Department of Botany, Charles University of Prague, 110 00 Prague, Czech Republic; 3Institute of Nature Conservation, Polish Academy of Sciences, 00-901 Krakow, Poland; 4Department of Botany and Biodiversity Research, University of Vienna, 1010 Vienna, Austria

**Keywords:** adaptation, alpine environment, ecotype, leaf anatomy, parallel evolution

## Abstract

Functional and structural adjustments of plants in response to environmental factors, including those occurring in alpine habitats, can result in transient acclimation, plastic phenotypic adjustments and/or heritable adaptation. To unravel repeatedly selected traits with potential adaptive advantage, we studied parallel (ecotypic) and non-parallel (regional) differentiation in leaf traits in alpine and foothill ecotypes of *Arabidopsis arenosa*. Leaves of plants from eight alpine and eight foothill populations, representing three independent alpine colonization events in different mountain ranges, were investigated by microscopy techniques after reciprocal transplantation. Most traits clearly differed between the foothill and the alpine ecotype, with plastic adjustments to the local environment. In alpine populations, leaves were thicker, with altered proportions of palisade and spongy parenchyma, and had fewer trichomes, and chloroplasts contained large starch grains with less stacked grana thylakoids compared to foothill populations. Geographical origin had no impact on most traits except for trichome and stomatal density on abaxial leaf surfaces. The strong parallel, heritable ecotypic differentiation in various leaf traits and the absence of regional effects suggests that most of the observed leaf traits are adaptive. These trait shifts may reflect general trends in the adaptation of leaf anatomy associated with the colonization of alpine habitats.

## 1. Introduction

Leaves are the main plant organs for photosynthetic carbon assimilation. Within a genetically fixed framework, leaf traits can be adjusted plastically to the conditions an individual plant experiences in order to optimize photosynthesis, respiration and transpiration [1,2]. Some adjustments can happen on a short time scale, within minutes, hours or days, and be reversible, e.g., leaf movements or changes in the orientation and ultrastructure of chloroplasts (transient acclimation) [2]. Other adjustments, such as plastic adjustments in leaf anatomical traits, occur over longer time periods, mirroring the environmental conditions during leaf formation and expansion, and are irreversible within the leaf lifespan [3]. Changes in morphological, anatomical, physiological and biochemical leaf traits in response to environmental conditions and the corresponding functional and ecological significance of these changes have received much attention [1,2,4,5,6]. For example, when exposed to high irradiance, leaves usually grow thicker with thicker palisade and spongy parenchyma and have well-developed cuticles and trichomes, thicker epidermal layers, tightly arranged spongy tissue, a higher stomatal density and an increased number of chloroplasts per leaf area with smaller grana but more stroma thylakoids [2,7,8]. However, low temperatures can also induce similar changes in leaf anatomy, whereby leaves of species from low-temperature zones typically are smaller with thicker epidermal cells and a thicker mesophyll as compared to leaves of species native to warmer-temperature zones [8]. Furthermore, plant growth under reduced partial pressure of atmospheric gases requires leaves to use CO_2_ with enhanced efficiency, which can be achieved by increased mesophyll thickness, accumulation of enzymes involved in photosynthetic carbon assimilation [9,10] and a high density of stomata [11]. Moreover, reduced water availability can lead to the formation of (more) trichomes, altered stomatal density and lowered cuticular conductance [12]. A reduced stomatal density on upper leaf surfaces prevents evapotranspiration, whereas more and smaller stomata on lower leaf surfaces enable more precise regulation of gas exchange [2,13]. In addition, the number of palisade layers often increases in response to water shortage, while the thickness of the spongy parenchyma and the volume of mesophyll cells and intercellular spaces decrease [14]. In summary, different abiotic stress factors can induce similar, or dissimilar, changes in leaf anatomy, challenging our understanding of the effects the environment has on leaf anatomy in natural habitats, in which plants experience ever-changing and interactive alterations in environmental conditions. Moreover, most studies on leaf anatomy have been conducted at broad phylogenetic scales comparing distantly related species [5,6], but tolerance of, and response to, environmental factors will vary between taxonomically distant plants, blurring the intrinsic relationships between leaf anatomy and environment [15]. Functional studies between closely related plant species or populations are still rare, as their phenotypes usually differ less clearly, making it more difficult to prove differences [16], but have the advantage of reducing potential bias resulting from different phylogenetic histories that influence trait expression, instead of environmental divergence. Furthermore, studies between closely related taxa allow sampling of related genetic information at high resolution and exploring the associated evolutionary mechanisms [17].

Plant species that form conspecific ecotypes in response to contrasting environmental conditions offer excellent opportunities for studying the significance of specific traits for plant existence in a given habitat in closely related and geographically closely located study systems bearing sufficient phenotypic variation. Ecotypes are locally adapted (groups of) populations within a species that are characterized by a combination of heritable and non-heritable traits [18,19] and are frequently still interfertile [20]. With the recent advance in genetic methods, ecotypes are receiving increasing interest [21], especially ecotypes resulting from parallel evolution [22], which is the independent polytopic evolution of ecotypes in response to similar selection pressures [23]. Parallelism in phenotypic traits in response to a particular set of environmental conditions strongly indicates that such traits are adaptive, i.e., that they have been selected for their contribution to enhanced fitness. In contrast, non-parallelism in trait expression may result from neutral processes such as genetic drift, or mirror environmental variation between local sites within a habitat. Conspecific ecotypes often occur in response to strong natural selection such as that imposed by extreme soil chemistry [24] but also occur along elevational gradients [25,26]. However, parallel evolution can be constrained by gene flow erasing ongoing differentiation, by variation in natural selection pressure counteracting differentiation or by small population size. As small populations often bear limited/low genetic variation, the source of adaptive variation is reduced, and genetic drift is increased due to a smaller number of individuals leading to stochastic changes [27,28]. However, not only particular traits *per se*, but also the ability to adjust them plastically can be under selection and differ between ecotypes [29,30].

Examples of parallel adaptation to different environments are known from various plant species, e.g., *Argyranthemum sundingii* [31], *Cerastium alpinum* [32], *Eucalyptus globulus* [33], *Heliosperma pusillum* [25], *Senecio lautus* [34], *Silene vulgaris* [35] and *Arabidopsis arenosa*. The latter, a biennial to perennial outcrosser, is an emerging model plant with distinct alpine ecotypes that arose *via* parallel evolution from foothill populations in several European mountain ranges [26]. *Arabidopsis arenosa* has a broad distribution over a wide range of environments from shaded rocks to dry steppes and sand dunes [36]. The considerable genomic variation between the distinct alpine and foothill ecotypes and among populations originating from geographically distinct mountain ranges [26] makes *A. arenosa* an excellent model for studying adjustment and adaptation of phenotypic traits. Furthermore, both diploid and tetraploid populations were found in one of these European mountain ranges, allowing studying the effects of ploidy [37].

A recent reciprocal transplantation study showed that differentiation between the alpine and the foothill *A. arenosa* ecotype is adaptive, e.g., related to the higher fitness of populations in their native *versus* foreign habitats [38]. Reciprocal transplantation experiments are often used to study adaptation mechanisms [28]. In contrast to *in situ* measurements of traits, they allow studying the extent of heritability and plasticity of phenotypic traits [2]. In a reciprocal transplantation design, populations of distinct ecotypes are grown in common gardens side by side in the natural habitats of both ecotypes, minimizing asymmetric bias of environmental conditions for one or the other ecotype. Grown under the same environmental conditions, genetically determined differences between populations remain, while environmentally induced plastic differences diminish [39]. Comparing traits of one ecotype in the native *versus* the foreign environment also allows concluding on its potential to plastically adjust a trait, which may be particularly important for phylogenetically young populations or ecotypes [29].

In the present study, we used the same reciprocal transplantation design as described by Wos et al. [38]. Eight alpine and eight foothill *A. arenosa* populations, originating from three European mountain ranges, the Niedere Tauern (Austria), the Făgăraș mountains (Romania) and the Tatra mountains (Slovakia), were grown in common gardens as previously described [38], comprising 12 tetraploid populations, and 4 diploid ones from the Tatra mountains, where both diploid and tetraploid populations occur. For these populations, it was previously shown [17,26] that a morphologically distinct alpine ecotype evolved from foothill populations independently in these mountain ranges, bearing significant footprints of parallel selection in multiple functionally relevant genes [17,26]. Individuals of the alpine ecotype are typically smaller, have fewer inflorescence stems bearing fewer but bigger flowers, flower later and for a shorter period, have a stricter requirement for winter vernalization, and typically invest more into vegetative than reproductive growth compared to those of the foothill ecotype [38,40]. We hypothesized that the parallel evolution of the alpine ecotype encompassed adaptation of leaf anatomy to environmental factors characterizing alpine habitats. We tested for parallel differentiation in leaf anatomical traits that could influence eco-physiological performance in the alpine or foothill habitat, which would strongly suggest that any such difference is adaptive. Furthermore, we compared the capacity of the alpine and foothill populations to plastically adjust leaf traits to the environmental conditions experienced in the alpine or foothill common garden. Finally, comparing diploid and tetraploid alpine and foothill populations, we evaluated whether polyploidization affected ecotypic differentiation.

## 2. Results

### 2.1. Environmental Conditions in the Common Gardens Used for Reciprocal Transplantation

Within the natural area of distribution of *A. arenosa*, an alpine common garden and a foothill common garden were established in the Niedere Tauern, Austria, as previously described [38]. The alpine common garden was established on Mt. Hohenwart at 2320 m a.s.l., and the foothill common garden was established in Aigen im Ennstal at 980 m a.s.l. (see Section 4 for further details), with clear differences in environmental conditions (Figure 1, Appendix A). The alpine common garden was snow-free later in the year and for shorter periods and characterized by much higher daily irradiation sums and irradiation maxima. The alpine common garden received around 1.3 and 3 times higher mean daily irradiation sums and on average about 50% and 70% higher mean values of daily maximum irradiance than the foothill common garden during summer 2018 and 2019, respectively (Appendix A). Daily mean leaf temperatures were lower in the alpine common garden, with 8.5 and 7.1 °C recorded during summer compared to 18.6 and 16.3 °C during the same time span in the foothill common garden in 2018 and 2019, respectively. The daily mean of relative humidity was comparable in both common gardens, but it was higher and fluctuated more in the alpine common garden (Figure 1, Appendix A). In summer 2018, 10 to 15% of the plants showed noticeable herbivory damage (as observed during the previously described plant phenotyping [38]), associated with the presence of caterpillars in the foothill common garden, whereas no such damage was observed in the alpine common garden.

### 2.2. Differences in Leaf Traits of Ecotypes Grown in the Alpine and Foothill Common Gardens

A PCA based on all measured leaf traits revealed that populations of the alpine and the foothill ecotype differed slightly depending on the common garden in which they grew, with a noticeable ecotypic differentiation within both native and foreign habitats. Alpine populations differed from foothill populations mainly in traits connected to leaf thickness (*x*-axis), i.e., total leaf thickness (LT), palisade parenchyma thickness (PT) and spongy parenchyma thickness (ST), although the shift was not very pronounced (Figure 2). However, when grown in the different common gardens, plants of the same ecotype split along the *y*-axis, mainly determined by trichome density (TD) and stomatal density (SD) (Figure 2).

A comparison of the individual leaf traits, using linear mixed models with ecotype, common garden, mountain range and their interaction as fixed factors (Figure 3, Appendix A), showed that most of the investigated traits, i.e., LT, PT and ST, and the relative proportion of palisade parenchyma (PP_rel_), relative proportion of spongy parenchyma (SP_rel_), adaxial trichome density (TD_ad_) and abaxial trichome density (TD_ab_) clearly differed between populations of the alpine and the foothill ecotype, with higher values in traits related to LT and lower values related to hairiness in alpine ecotypes (Figure 3, Appendix A). Figure 4 shows selected examples of leaf cross sections. Other traits, such as specific leaf area (SLA), adaxial and abaxial epidermal thickness (ET_ad_ and ET_ab_, respectively), and abaxial stomatal density (SD_ab_) only differed between ecotypes depending on the common garden (Appendix A, ecotype: common garden interaction in Appendix A).

### 2.3. Ecotype Plasticity

Alpine and foothill ecotypes had a different potential to adjust most of their leaf traits (LT, ST, ET_ad_, ET_ab_, PP_rel_, SP_rel_, SLA and SD_ab_) plastically to the environmental conditions experienced in the common garden in which they grew (see ecotype: common garden interactions in Appendix A). The evolutionary origin of the individuals studied only influenced SD_ad_ and TD_ad_ (factor region in Appendix A) and did not substantially influence the ecotypic differentiation for any of the measured traits (interaction ecotype: region in Appendix A). The extent of plasticity, as depicted by the logarithm of the ratio of trait values for individuals of the same population of origin grown in their native versus foreign habitat, only differed for ST, ET_ad_, SLA and SD_ab_ (Appendix A), and ST and ET_ad_ were more plastic in the alpine ecotype (Appendix A). Instead, SLA and SD_ab_ were more plastic in the foothill ecotype (Appendix A).

### 2.4. The Effect of Ploidy on Leaf Traits

When diploid and tetraploid individuals from the Tatra mountains were compared, the overall pattern observed was similar to that of tetraploid populations, with substantial influences of the local environment (common garden), and to some extent of the ecotype, on many leaf traits (Appendix A). Ploidy level only had a significant influence on SD on both leaf sides, which was greater in diploid populations (Appendix A). The interaction of ecotype and ploidy was not significant for any of the leaf traits studied. No differences in the extent of plasticity were detected for any leaf trait, except for SD_ad_, which was more plastic in tetraploid than diploid populations (Appendix A).

### 2.5. Chloroplast Ultrastructure

Chloroplast ultrastructure was comparable within individuals of each ecotype but differed greatly between individuals of the alpine and the foothill ecotype, as illustrated with a representative example in Figure 5. When plants were grown together in the alpine common garden, chloroplasts of the foothill ecotype were lens-shaped and almost without starch and had higher stacked grana thylakoids (Figure 5a). In contrast, chloroplasts of the alpine ecotype were roundish and filled with massive starch grains and had less stacked grana thylakoids (Figure 5b). Both ecotypes contained plastoglobules, but they were larger and more abundant in the foothill ecotype (Figure 5).

## 3. Discussion

Plants migrating from lower elevations into alpine habitats experience concerted changes in multiple environmental factors. Partial pressure of atmospheric gases and temperature decrease with increasing elevation, while precipitation increases [1]. Alpine habitats are also characterized by higher daily and seasonal fluctuations in environmental factors such as temperature and irradiation, and local variation in environmental conditions is usually higher than that in foothill habitats [1,41,42]. In agreement with the environmental differences between the foothill and the alpine habitats consistently observed in all three mountain ranges in which the populations evolved [26], most leaf anatomical traits differed heritably between individuals of the alpine and foothill ecotypes, irrespective of their independent evolutionary origin (Figure 3, Appendix A). The two ecotypes also differed in their capacity to adjust their leaf anatomy plastically to the environmental conditions experienced in the common garden in which they grew (Appendix A). The parallel differentiation evidenced for most leaf traits in populations from all three mountain ranges strongly suggests that those traits were selected for conferring a fitness advantage to each ecotype at its elevation of origin, in line with the previous evidence of local adaptation in *A. arenosa* [38]. Conversely, neutral differentiation would lead to stochastic, non-parallel differences between populations evolving within different mountain ranges. Although the alpine ecotype has evolved several times independently in at least three mountain ranges [26], non-parallel patterns in the traits measured were surprisingly rare (Appendix A).

A previous study of differences in morphological traits of plants of both ecotypes grown in their natural habitats also revealed a high extent of parallelism added to non-parallelism in some traits that often disappeared when individuals were grown under the same conditions, suggesting that phenotypic plasticity was the likely major driver of that non-parallelism [26]. The comparison of plants from the same populations grown in each of the two common gardens revealed plastic adjustments of various leaf anatomical traits, and the extent of plasticity differed between the two ecotypes (Figure 3, Appendix A). As compared to the foothill ecotype, the evolutionarily younger alpine ecotype showed greater plasticity for traits related to leaf thickness when grown in the alpine common garden, adding to the constitutive ecotypic differences found for those traits. Instead, SLA and SD_ab_ were more plastic in the foothill ecotype (Appendix A). Although the direction of change (e.g., a greater LT in leaves grown in the alpine common garden) was shown to be adaptive in earlier studies of alpine species, it remains unclear whether a higher plasticity alone (Appendix A) represents a fitness advantage [43]. It is often assumed that phenotypic plasticity frequently evolved as an adaptation to environmental heterogeneity, being especially important for sessile organisms. However, phenotypic responses may also result from genetic correlations with other traits that are under selection or due to genetic drift [30]. Furthermore, high levels of phenotypic plasticity can also be inherently costly, and thus maladaptive [44,45]. Indeed, adaptive, maladaptive and neutral consequences of plasticity have been reported [46,47,48,49,50]. The role of phenotypic plasticity in the course of evolution is also under debate. On the one hand, phenotypic plasticity may reduce the power of natural selection, hindering specialization *via* genetic differentiation [51]. On the other hand, enhanced plasticity may enable the colonization of new environments, fostering adaptation [29]. *Arabidopsis arenosa* represents an excellent system to test for the role of plasticity in response to habitat differentiation, and future studies could test for adaptive advantages, or disadvantages, of the extent of plasticity.

Overall, individuals of the alpine ecotype developed thicker leaves (LT), with thicker palisade (PT, PP_rel_) and spongy parenchyma (ST, SP_rel_) than those of the foothill ecotype, and this difference was most evident in plants grown in the alpine common garden, where epidermal layers (ET_ad_, ET_ab_) were also thicker (Figure 3, Figure 4 and Appendix A). This observation is consistent with general trends for alpine species having thicker leaves than lowland species as an adaptation to higher irradiance in the alpine habitat [1]. A thicker palisade parenchyma, containing the majority of chloroplasts and channeling direct light, enables plants to increase net photosynthetic rates [52,53,54,55]. The thicker mesophyll can also allow increasing the surface of chloroplasts facing intercellular spaces, thereby favouring CO_2_ uptake into the cells and hence compensating for the lower CO_2_ pressure at increased elevation [1,54,56]. Congruently, higher net photosynthetic rates under high irradiances were measured in the alpine ecotype grown in the alpine common garden as compared to the foothill ecotype (Kaplenig et al. unpublished). Moreover, differences in leaf ultrastructure indicative of adjustment to higher irradiation were visible in chloroplasts of the alpine ecotype grown in an alpine common garden (Figure 5). While the chloroplasts of the alpine ecotype contained several starch grains indicative of good photosynthetic performance, the chloroplasts of the foothill ecotype were virtually free of starch grains, and the plastoglobules were larger and more abundant, indicative of light stress-dependent re-arrangements of thylakoid membranes [57] in the foothill ecotype when grown in the alpine common garden (Figure 5). Grana stack thickness was increased in the foothill ecotype compared to the alpine one, but it should be noted that this observation cannot be directly linked to physiological performance, as structural and organizational changes in thylakoid structure are driven by physico-chemical forces leading to dynamic flexibility [58]. The increased leaf thickness in plants from the alpine common garden could be further influenced by the lower temperatures in the alpine environment, which can also increase the thickness of epidermal cells and decrease leaf area [8].

In contrast to their alpine counterparts, leaves of foothill plants had higher TDs on both leaf surfaces regardless of the common garden in which they grew (Figure 3; Appendix A). Trichomes can serve as protection against multiple abiotic and biotic stress factors [59]. Higher TDs can elevate the air humidity on the leaf surface and boundary layer resistance, thereby reducing transpiration, but at the expense of CO_2_ diffusion. Thus, higher TD may be beneficial under the drier conditions observed in the foothill common garden (Figure 1, Appendix A, [26]), but it is presumably not advantageous under lower CO_2_ partial pressure. Furthermore, trichomes may protect from herbivores [60]. As herbivore pressure tends to decrease with increasing elevation [61], having lower TD may be cost-efficient for alpine plants [62]. Reduced constitutive chemical and morphological defence against herbivores was observed in plants growing at higher elevations [61,63,64,65], which were also found to be more palatable to generalist herbivores than those growing at lower elevations [62,66,67,68,69]. In agreement with these studies, the observed higher occurrence of herbivory damage in plants growing in the foothill common garden as compared to the alpine one supports the assumption that in *A. arenosa*, higher TD was selected as a defence mechanism against herbivores at lower elevations. Conversely, investing in higher trichome density may not represent a cost-efficient strategy for alpine *A. arenosa* populations.

The patterns of ecotypic differentiation for the measured leaf anatomical traits were highly congruent in diploid and tetraploid populations (Appendix A). Indeed, the difference in ploidy did not have a significant effect on trait expression, except for SD on both leaf surfaces, which was higher in diploid populations of both ecotypes as compared to tetraploid ones. As stomatal number and size are usually inversely correlated [70], SD frequently changes as a direct consequence of whole genome duplication, because polyploidization leads to an increase in cell size, also affecting stomatal guard cells [71,72,73,74,75]. Therefore, lower SD in tetraploid plants as compared to diploid ones may be accompanied by larger stomatal guard cells [76]. Larger stomata result in a lower diffusive resistance to CO_2_, and thus, CO_2_ gas exchange can be enhanced with ploidy [74]. However, having fewer but bigger guard cells could be disadvantageous in habitats where soil moisture is not always sufficient, such as the habitat of the foothill ecotype, which can be occasionally very dry (Figure 1, Appendix A, [26]). In the course of evolution, (maladaptive) traits may be adjusted, e.g., a reduction in cell size can be initiated because fewer and smaller stomata could allow a more precise regulation of gas exchange [2].

In summary, albeit receiving much attention historically, our understanding of the environmental effects on leaf anatomy can be confounded by overlaps and interaction between the effects of individual environmental stress factors and the use of taxonomically distant plant species. In view of the pivotal role of leaf traits to plant function, changes in leaf traits in response to the environment deserve revisitation with contemporary approaches. In this study, we used the opportunity offered through conspecific ecotypes of the emerging new model species *A. arenosa* to show that leaf anatomical traits have changed with remarkable consistency across three different mountain ranges. Taken together, this parallel, heritable differentiation between the foothill and the alpine ecotype regarding various leaf traits and the absence of regional effects strongly suggests that most of the differences in leaf traits are adaptive. The parallel evolution leading to consistent trait shifts in leaf anatomy towards thicker, less hairy leaves with thicker palisade and spongy parenchyma in the alpine ecotype supports the hypothesis that these leaf traits are pivotal parts of the suite of plant traits that define the alpine ecotype in *A. arenosa*. The example of *A. arenosa* illustrates that plant species often bear significant intraspecific variation in anatomical, morphological and functional traits in relation to distinct habitats. Therefore, conservation of biodiversity and related ecosystem functions requires conserving genomic diversity including ecotypes and not only species [77], an aspect that is particularly relevant in the face of the current threats posed to biodiversity by environmental change.

## 4. Materials and Methods

### 4.1. Experimental Design and Biological Material

Anatomical traits were studied in leaves of *Arabidopsis arenosa* (L.) Lawalrée plants from alpine and foothill populations grown in a previously described reciprocal transplantation experiment [38,78]. Briefly, plants were grown from seeds collected in 2014 from at least 10 mother plants each from 16 populations originating from three mountain ranges. These were the Niedere Tauern (Eastern Alps, Austria, NT), the Tatra Mountains (Western Carpathians, Slovakia, and populations came from two sub-ranges, Vysoké Tatry (VT) and Západné Tatry (ZT)) and Făgăraș (Southern Carpathians, Romania, FG). In all of these mountain ranges, an alpine ecotype evolved in parallel from a foothill ecotype [38]. Two tetraploid alpine populations and two tetraploid foothill populations from each mountain range were studied. Additionally, two diploid foothill populations and two diploid alpine populations from the Tatra Mountains, where both diploid (VT) and tetraploid populations (ZT) occur, were used. The first generation of plants was grown under controlled conditions in growth chambers as described by Wos et al. [79]. Mature seeds were harvested and stored at 4 °C in darkness under dry conditions. In spring 2018, seeds of the second generation were stratified at 4 °C for 4 days prior to germination at 20 °C under constant light. Seedlings were transplanted into multipot trays, grown in a greenhouse under ambient light and temperature conditions with minimum temperatures above 8 °C, and transplanted to common gardens after they had developed at least 4 leaves. Two common gardens were established within the natural habitats of a foothill and an alpine population of *A. arenosa* in Styria, Austria (Niedere Tauern), one at 980 m a.s.l. in Aigen im Ennstal and the other at 2320 m a.s.l. on the northern slope of Mt. Hohenwart. At each site, a minimum of 830 plants were randomly transplanted into 10 × 10 cm cells of 1 × 1 m plots in natural soil.

Additionally, for transmission electron microscopy, a third generation of seeds from all 16 populations was produced by growing plants potted into soil containing a mixture of “alpine soil” (consisting of leaf mold, topsoil, Lavalit, peat, sand and rock meal (5:2:1:2:2:0.2)), silicate sand and vermiculite (8:1:1) in a greenhouse located in the Botanical Garden of the University of Innsbruck under ambient light and temperature conditions from June 2018 until seed collection between April and June 2019. Seeds were equilibrated at 30% relative humidity and stored at 23 °C in hermetic containers. Third-generation seeds were germinated at the end of July 2019 as described above and transplanted to individual pots containing natural soil collected at the common garden on Mt. Hohenwart and the soil mixture used for seed production (2:1). Pots were then buried in sand in a common garden on Mt. Patscherkofel (close to Innsbruck) at the beginning of September 2019 and watered as required, until sampling in June 2020.

### 4.2. Microclimate

Microclimatic data were recorded at 6 min intervals with climate stations (CR1000, Campbell Scientific, Logan, UT, USA) powered by solar panels as previously described [78]. Photosynthetic photon flux density (PPFD) was measured with quantum sensors (QS; SKP 215, Skye Instruments Ltd., Wales, UK), air temperature and relative humidity with temperature and humidity sensors (HC2S3, Campbell Scientific, Logan, UT, USA) and leaf temperature by copper-constant thermocouples placed on the lower leaf surfaces of at least 10 individuals per common garden with a gas-permeable tape (3M Transpore, North Coast Medical Inc., Morgan Hill, CA, USA). Soil temperature was recorded using a thermocouple probe 10 cm below the surface (Probe 107; Campbell Scientific).

### 4.3. Sampling and Leaf Anatomy

In each common garden, individuals with at least five fully developed rosette leaves were sampled. At five (Aigen) and two (Mt. Hohenwart) sampling dates, plants were transplanted into pots using soil from the respective sites (Appendix A). Plants were watered and within one day transferred to a climate chamber (Percival PGC-6HO, Percival Scientific Inc., Iowa, USA) at the Department of Botany in Innsbruck for the period of investigation, i.e., for three to four days. Temperature was set to the average temperature recorded on leaf rosettes during the week preceding the sampling at the respective common garden, using the current daylength with an irradiation intensity adjusted to 800 µmol photons m^−2^ s^−1^ (Appendix A).

Fully developed rosette leaves collected from three to seven individuals per population and common garden were used for assessing leaf anatomy. Leaf cross sections were made from the middle section of leaves using a hand microtome (GSL1, Schenkung Dapples, Zürich, Schweiz) at an angle of 0°, 45° and 90°. From each individual plant, five cross sections were made in each cutting direction, from which one was randomly chosen for analyses. Leaf cross sections were investigated with a light microscope (Olympus BX50F, Tokyo, Japan) equipped with a camera (Olympus DP25, Tokyo, Japan). Anatomical measurements were conducted on digital images using the software CellD and CellSens (CellD Version 3.1 and CellSens Entry 2.3, Olympus Optical Co., Tokyo, Japan).

The leaf thickness (LT), palisade parenchyma thickness (PT), spongy parenchyma thickness (ST) and epidermis thickness on adaxial (ET_ad_) and abaxial (ET_ab_) surfaces were determined using three cross sections per individual plant, using the mean of five measurements per cross section. The relative proportions of palisade parenchyma (PP_rel_), spongy parenchyma (SP_rel_) and intercellular spaces (IC_rel_) were estimated from one image per cross section per individual (three images per individual) according to Kubinova [80]. Stomatal densities of adaxial (SD_ad_) and abaxial (SD_ab_) leaf surfaces were determined from imprints. For imprints, nitrocellulose polish (Power Pro Nail Lacquer, KIKO Milano S.p.A, Bergamo, Italy) was applied to whole leaves as a thin film and carefully removed after 5 min of air drying using transparent adhesive tape, which was then placed on microscope slides. Stomatal density was determined from digital microscopic images of imprints as the mean number of stomata located within five randomly chosen grid cells of 120 × 120 μm in the middle section of leaves. Trichome densities of adaxial (TD_ad_) and abaxial (TD_ad_) surfaces were determined on digital images of leaf surfaces obtained by the use of a reflected-light microscope (Olympus SZX12, Olympus Optical Co., Tokyo, Japan) with a camera (Olympus C-7070, Olympus Optical Co., Tokyo, Japan). The number of trichomes was counted using the software ImageJ [81] and averaged to an area of 1 mm^2^. Specific leaf area, the ratio of leaf area to dry weight, was determined from one well-developed leaf per individual. Leaves were placed together with a millimetre scale under a microscope (Olympus SZX12, Olympus Optical Co., Shinijuku, Japan) and photographed with a camera (Olympus C-7070, Olympus Optical Co., Tokyo, Japan). Leaf area was determined from pictures using ImageJ [81]. Photographed leaves were then dried for about one week at 65 °C in an oven (T 6060, Heraeus, Hanau, Germany) and thereafter transferred to a desiccator above silica gel beads and weighed with a precision balance (Sartorius Lab Instruments GmbH & Co., KG, Göttingen, Germany).

In addition, leaves collected from plants that were grown on Mt. Patscherkofel were investigated by transmission electron microscopy using adult, healthy rosette leaves of the same developmental stage sampled on a sunny day around noon. Leaf samples were immediately fixed in 2.5% glutaraldehyde in 50 mM sodium cacodylate buffer, pH 7.0, for 2 h and rinsed with sodium cacodylate buffer. Samples were post-fixed in buffered 1% OsO_4_ at 4 °C overnight and further processed as previously described [82]. Ultrathin sections were stained with 2% uranyl acetate and lead citrate and investigated with a Zeiss Libra 120 TEM (Carl Zeiss, Oberkochen, Germany) at 80 kV. Images were taken with a 2 × 2 k digital high-speed camera (Tröndle, Moorenweis, Germany) under the control of ImageSP software (Tröndle, Moorenweis, Germany).

### 4.4. Statistical Analysis

All analyses were performed using the software “R” at a significance level of *p* < 0.05 [83]. Principal component analysis (PCA) was applied to illustrate differences among ecotypes and common gardens using the leaf anatomical traits, LT, PT, ST, ET_ad_, ET_ab_, SD_ad_, SD_ab_, SLA, TD_ad_ and TD_ab_, standardized to zero mean and unit variance by the use of the package ade4 [84]. Missing values (in rare cases when trichomes or stomata could not be accurately determined from the surface imprints) for some individuals were replaced by the group mean. As the Spearman correlation index was lower than 0.8 for any pair of traits, all traits were retained. To test for differences in leaf traits, linear mixed models (LMEs), as implemented in the function “lmer” in the package “lmerTest” [85], were applied separately for each trait. Source population was used as a random factor to account for the potential non-independence of values derived from seeds sampled within the same population. We first used models to test for ecotypic differentiation and plastic adjustment in leaf traits between tetraploid foothill and alpine populations, and parallel evolution of the latter in the three mountain ranges: ecotype, common garden (foothill vs. alpine) and mountain range as well as the interactions ecotype–common garden and ecotype–mountain range were regressed as fixed effects on each leaf trait. Through this approach, ecotypic differences are characterized by a significant effect of the factor ecotype or a significant interaction of the factors ecotype and common garden. Plastic adjustments of traits to the common garden are depicted by significant interactions of the factors ecotype and common garden. A significant effect of region indicates non-parallelism between populations of one ecotype. Parameters were estimated by optimizing the restricted maximum likelihood criterion. Another set of models was applied to all leaf traits to test for the influence of ploidy on ecotypic differentiation, using ecotype, ploidy and common garden as well as the interaction ecotype–ploidy and ploidy–common garden as fixed effects and population as a random factor. Only data from the Tatra mountains were used in this approach, as diploids occur only in this mountain range. We checked the model assumptions of normality and homogeneity of error variance by diagnostic plots, calculating variance inflation factors (VIFs) and testing for homogeneity of variance by Levene’s test. Traits were transformed by natural logarithm (LT, PT, ET_ad_, ET_ab_, SLA), square root (ST, IC_rel_, SD_ad_, SD_ab_) and cube root (TD_ad_, TD_ab_) for LMEs and PCA. For leaf traits assessed in three cutting directions, a random factor was added to the model to account for the nesting of traits within the individual.

The extent of plasticity in responses to environmental conditions was assessed by comparing leaf traits of plants originating from the same population but transplanted to alpine and foothill common gardens. As we did not work with clones, but individuals raised from seeds, we applied a random assignment of individuals transplanted in the alpine common garden to those growing in the foothill common garden. The assignment was repeated 100 times, and for each pair of individuals, the logarithms of the ratios (alpine/foothill) for each leaf trait were calculated and then compared using linear mixed models, with ecotype and mountain range as fixed factors and population as a random factor.

## Figures and Tables

**Figure 1 plants-11-02626-f001:**
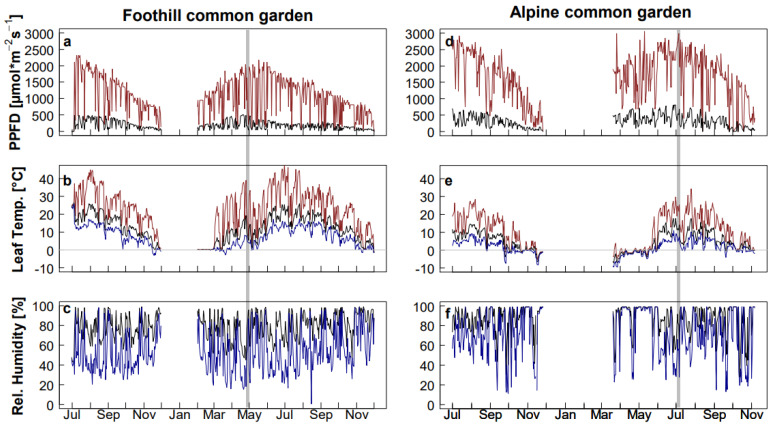
Environmental conditions in the common gardens used for reciprocal transplantation. Data were collected between July 2018 and November 2019, except for time periods with snow coverage, in common gardens set up in the natural habitats of (**a**–**c**) the foothill and (**d**–**f**) the alpine ecotype. (**a**,**d**) Daily maxima (red) and means (black) of photosynthetic photon flux density (PPFD). (**b**,**e**) Daily maxima (red), means (black) and minima (blue) of leaf temperature, averaged for rosettes of six individual plants each; thin grey horizontal lines in (**b**,**e**) indicate 0 °C. (**c**,**f**) Daily means (black) and minima (blue) of relative humidity. Grey bars show the sampling dates at the onset of the growing season in the respective habitat.

**Figure 2 plants-11-02626-f002:**
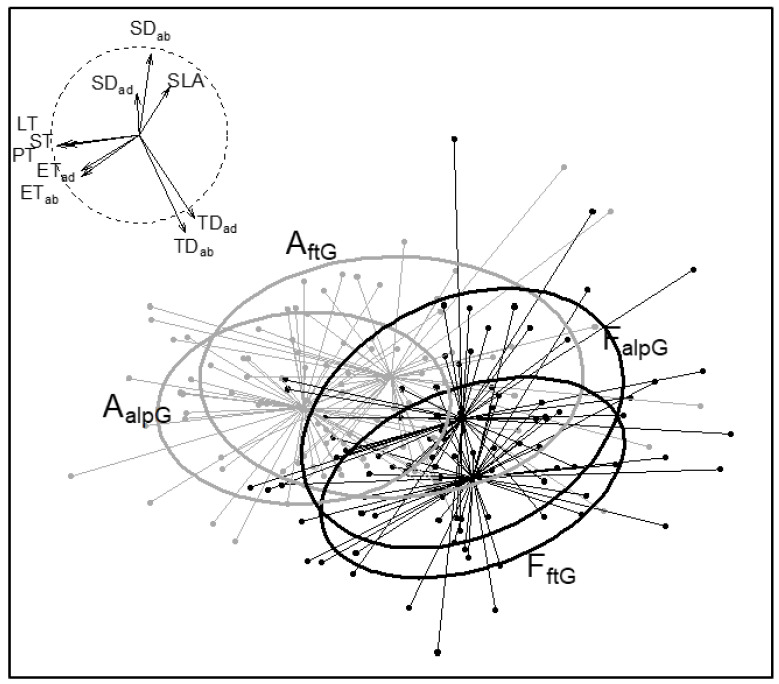
Effects of ecotype and environment on leaf anatomy in *A. arenosa*. Individuals from alpine (A, grey) and foothill (F, black) *A. arenosa* populations were grown in an alpine common garden (alpG) and a foothill common garden (ftG). Principal component analysis was conducted using leaf anatomical traits, i.e., total leaf thickness (LT), palisade parenchyma thickness (PT), spongy parenchyma thickness (ST), mean thickness of the epidermis on adaxal (ET_ad_) and abaxial (ET_ab_) leaf surfaces and mean stomatal density on adaxial (SD_ad_) and abaxial (SD_ab_) leaf surfaces. Variables were transformed to achieve normal distribution, scaled and centered. Confidence ellipses were defined by the centroid and SD of the cloud. Axes represent 39.4% (*x*-axis) and 16.3% (*y*-axis) of the explained variance. Inset: arrows in the dashed circle were obtained by a second projection and indicate the contribution of traits to the separation along the principal components.

**Figure 3 plants-11-02626-f003:**
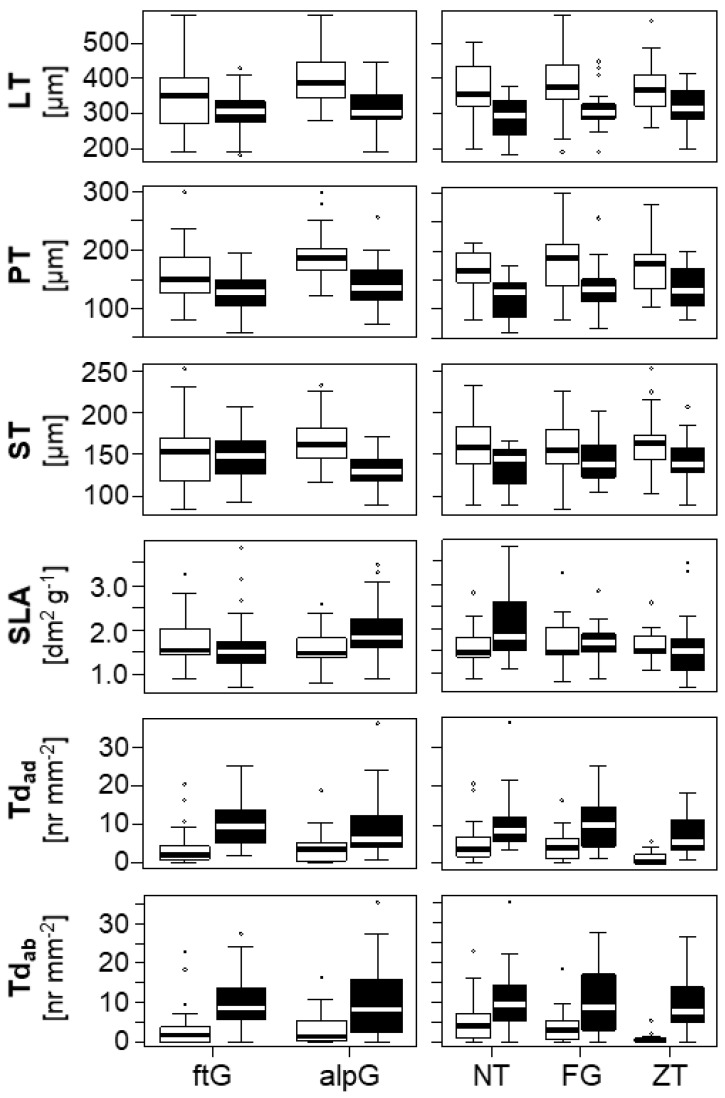
Anatomical traits of leaves of tetraploid *A. arenosa* populations originating from three mountain ranges, i.e., the Niedere Tauern (NT), the Făgăraș mountains (FG) and the Tatra mountains (ZT). Leaf anatomical traits shown are total leaf thickness (LT), palisade parenchyma thickness (PT), spongy parenchyma thickness (ST), specific leaf area (SLA) and trichome density per area on adaxial (TD_ad_) and abaxial (TD_ab_) leaf surfaces. White and black bars denote alpine and foothill populations, respectively, grown in a foothill (ftG) and an alpine (alpG) common garden. Box plots show medians and the 25th and 75th percentiles, and dots outside 1.5× interquartile ranges represent outliers. Data for all tetraploid populations per ecotype were pooled; panels in the left row show comparisons between the two different common gardens, and panels in the right row show comparisons across mountain ranges.

**Figure 4 plants-11-02626-f004:**
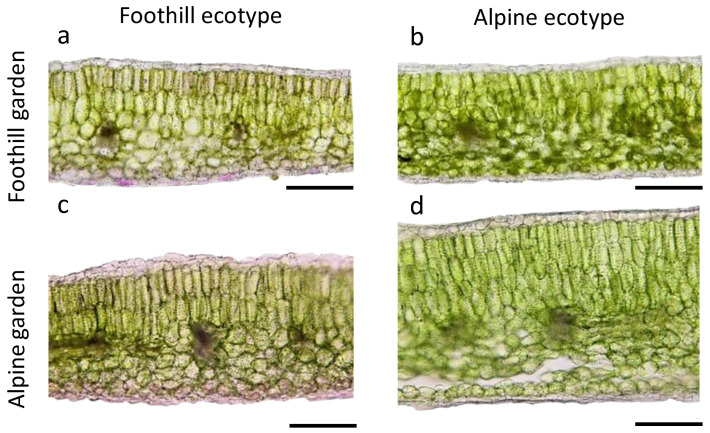
Ecotypic differences in leaf anatomy of plants from alpine and foothill *A. arenosa* populations submitted to a reciprocal transplantation experiment, visualized by light microscopy. Representative examples of leaf cross sections are shown for plants from (**a**,**c**) a foothill population and (**b**,**d**) an alpine population that were grown together in (**a**,**b**) a foothill and (**c**,**d**) an alpine common garden. Scale bars: 200 µm.

**Figure 5 plants-11-02626-f005:**
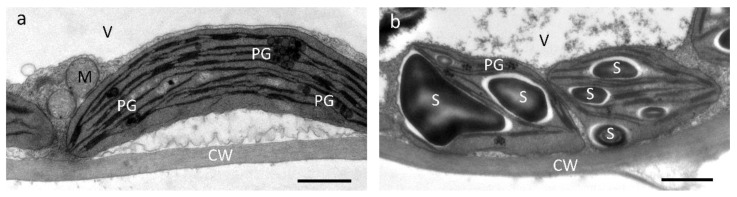
Ecotypic differences in chloroplast ultrastructure in mesophyll cells of *Arabidopsis arenosa* plants shown by transmission electron microscopy. Representative examples are shown for plants from (**a**) a foothill population and (**b**) an alpine population grown together in an alpine common garden. Letters indicate the cell wall (CW), plastoglobules (PG), mitochondrion (M), starch grain (S) and vacuole (V). Scale bars: 1 μm.

## Data Availability

Not applicable.

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
