# Peer review of "Parallel Differentiation and Plastic Adjustment of Leaf Anatomy in Alpine Arabidopsis arenosa Ecotypes"

_plants, 2022, doi:10.3390/plants11192626_

Round 1
Reviewer 1 Report
The article is well written, divided into easy-to-read sections, and supported by adequate citations. I have no concerns over the title, the content of the presentation, or the references presented.
However, several statements are repeated throughout the introduction section, this has to be avoided to
make the paper more comprehensive.
Please, briefly add future perspectives and further applied applications of this specific research work in the discussion section or conclusion section.
Author Response
Our Response:
We eliminated redundancy in the introduction and added a short section on perspectives and applications at the end of the Discussion.
The changes in the manuscript are displayed in blue font.

Reviewer 2 Report
The revised paper is dedicated to a very interesting topic of distinguishing between genetic and environmental components of variability, i.e. the problem as old and still actual as the whole genetics. This manuscript is well written and illustrated, so I can recommend to publish this work in Plants.
I have some minor concerns mostly about style and, in several minute aspects, language (see attached manuscript file with corrections and comments).
As for more serious suggestions, I would recommend authors to make the statistical part(s) more concise. There are statements that some features differed between specimens/trials 'clearly' or 'less markedly' (subsection 2.2), but only statistical significance is important. It is necessary to add any indicators of significance of differences in all boxplots (Fig. 3 and supplement).
After taking suggestions into account, the reviewed work can be recommeded for publication.

Author Response
Reviewer 2: The revised paper is dedicated to a very interesting topic of distinguishing between genetic and environmental components of variability, i.e. the problem as old and still actual as the whole genetics. This manuscript is well written and illustrated, so I can recommend to publish this work in Plants. I have some minor concerns mostly about style and, in several minute aspects, language (see attached manuscript file with corrections and comments). As for more serious suggestions, I would recommend authors to make the statistical part(s) more concise. There are statements that some features differed between specimens/trials 'clearly' or 'less markedly' (subsection 2.2), but only statistical significance is important. It is necessary to add any indicators of significance of differences in all boxplots (Fig. 3 and supplement). After taking suggestions into account, the reviewed work can be recommeded for publication.
We agree with all comments of reviewer 2 and amended the manuscript accordingly (displayed in blue font in the text), with the following exceptions:
We prefer to keep the term “mountain range”. We believe that this term is appropriate (a mountain range can comprise several mountain ridges, which is the case for the mountains described in this study) and also want to be consistent with earlier studies conducted with the same biological material from the same mountains (Kaplenig et al. 2022, Wos et al. 2022).
In the abstract, “large starch grains” is correct, not “larger starch grains”, as the alpine ecotype had large starch grains, whereas the foothill one had no starch grains at al.
It seems that it was not clear that all statistical details are provided in the Supplementary Material. Showing asterisks in the boxplots would not be compatible with two-factorial linear mixed models. All statements “clearly differed” and “differed less markedly” referred to statistically significant differences; the latter term has been removed nonetheless (we wanted to stress the amplitude of the differences, e.g. although statistically significant, small differences may be less meaningful in a biological sense). However, in some cases, references to the Supplementary Tables were missing and these are now included.
Finally, we appreciate the reviewer’s comment on plasticity! We are aware that our test for differences in the extent of plasticity does not allow to distinguish between the plasticity of single individuals (genotypes) and genomic variability (potentially available genomic polymorphism) within a population. Both, increased plasticity and genomic variability could be selected for in variable environments. This aspect would be an interesting and promising aspect for further studies in the model plant, but this was not within the remits of this study.
